# Considerations for data acquisition and modeling strategies: Mitosis detection in computational pathology

**Zongliang Ji**[1]                                          JERRYJI@CS.TORONTO.EDU
**Philip Rosenfield**[1]                              PHILIP.ROSENFIELD@MICROSOFT.COM
**Christina H Eng**[2]                                       CENG@VOLASTRAX.COM
**Sarah E Bettigole**[2]                                 SBETTIGOLE@VOLASTRAX.COM
**Danielle C Gibson**[3]                                  DANIELLE@HISTOWIZ.COM
**Hamid Masoudi**[4]                          HMASOUDI@PROVIDENCEHEALTH.BC.CA
**Matthew G Hanna**[5]                                      HANNAM@MSKCC.ORG
**Nicolo Fusi**[1]                                         FUSI@MICROSOFT.COM
**Kristen A Severson**[1]                             KSEVERSON@MICROSOFT.COM

[1] *Microsoft Research, New England*

[2] *Volastra Therapeutics*

[3] *Histowiz*

[4] *Department of Pathology, St. Paul's Hospital, University of British Columbia*

[5] *Department of Pathology, Memorial Sloan Kettering Cancer Center*

**Editors:** Accepted for publication at MIDL 2023

## Abstract

Preparing data for machine learning tasks in health and life science applications requires decisions that affect the cost, model properties and performance. In this work, we study the implication of data collection strategies, focusing on a case study of mitosis detection. Specifically, we investigate the use of expert and crowd-sourced labelers, the impact of aggregated vs single labels, and the framing of the problem as either classification or object detection. Our results demonstrate the value of crowd-sourced labels, importance of uncertainty quantification, and utility of negative samples.

**Keywords:** Computational pathology, mitosis detection, breast cancer

## 1. Introduction

The application of computer vision techniques and models for analysis of medical images is compelling because of the potential to generate fast, accurate, and reliable predictions, leading to applications which would benefit both patients and clinicians. Although there is growing excitement around the use of self-supervised methods to decrease the dependence of machine learning (ML) models on annotated datasets (see recent surveys by Ciga et al., 2022; Krishnan et al., 2022), the primary paradigm for using ML models in medical applications is supervised learning. Indeed, much of the literature focuses on scenarios where the input data is pre-collected and fixed without regard to the data gathering phase.

In this work, we seek to investigate the interactions between label acquisition and modeling choices. Specifically we look at trade-offs in the cost of label acquisition vs model performance by comparing the performance of classification models and object detection models. For label acquisition, classification models have lower costs, as they can be trained

using comparatively easy-to-acquire binary labels. Conversely, object detection models have higher costs in label acquisition, as they require detailed annotations. We also consider the impact on performance of different labeling schemes. We compare using annotations from only one labeler to annotations from multiple labelers (per input). We focus on the application of mitosis detection in digital pathology images, but note that the characteristics of the problem can be abstracted and applied to other settings.

Mitosis detection, the identification of cells undergoing cell division, is a well-studied problem as it is an important metric for tumor proliferation in cancer. In the context of clinical practice, a simplified detection pipeline is as follows: during cancer diagnosis, a tumor tissue sample is acquired, stained using hematoxylin and eosin (H&E), and then assessed by a pathologist who makes a diagnosis. The pathologist will count mitotic events (cells observed to be undergoing mitosis) during analysis of the stained tissue under a microscope. Generally speaking, more mitotic events indicates greater proliferation speed in cancer tumors.

In recent years, these tissue samples are increasingly being digitized, enabling *computational pathology* (Fuchs and Buhmann, 2011; Cui and Zhang, 2021; Srinidhi et al., 2021), a domain broadly referring to the use of computational tools for pathology tasks. Because of the success of machine learning models in natural images, there is excitement around the potential of ML models for pathology tasks, which could help speed-up prediction time, reduce pathologist burden and perhaps increase accuracy and reliability in predictions (Abels et al., 2019; Van der Laak et al., 2021; Rajpurkar et al., 2022; Javed et al., 2022).

Mitosis detection is a natural setting for our analysis. The problem characteristics admit formulations as either classification or object detection. This is because digital histopathology images are very large, up to 150k × 150k pixels, and therefore the first step of a computational histopathology pipeline is to divide the image into smaller sub-images, called tiles. Depending on the size of these tiles, it may be sufficient to associate labels with the tile-level thus mapping naturally to a classification problem. Alternatively, labels may be bounding boxes on the tile, mapping naturally to an object detection problem. Exploiting this duality, there have been several machine learning challenges in mitosis detection which use bounding boxes (or centroids) such as as MITOS 2012 (Ludovic et al., 2013) and TUPAC16 (Veta et al., 2019), and tile-level annotations, such as AMIDA13 (Veta et al., 2015). There have also been studies which attempt to merge both types of labels to increase dataset size (Mehta et al., 2018; Sebai et al., 2020a; Ciga and Martel, 2021).

Mitosis is a temporal process and therefore has distinct visual features depending on the stage of cell division. This complexity is further compounded by non-mitotic events, such as programmed cell death, which can appear very similarly to mitosis, particularly to the untrained eye. Therefore, mitotic event labels are typically the consensus of two or more labelers, motivating the additional aspects of our analysis.

## 2. Background and related work

This study explores trade-offs in data acquisition and modeling choices for mitosis detection in histopathology images. To investigate, we created a pipeline beginning with raw histopathology images of a patient's biopsy (SVS formatted files, hereafter, whole slide images; WSI) collected by The Cancer Genome Atlas BReast CAncer (TCGA BRCA) pro-

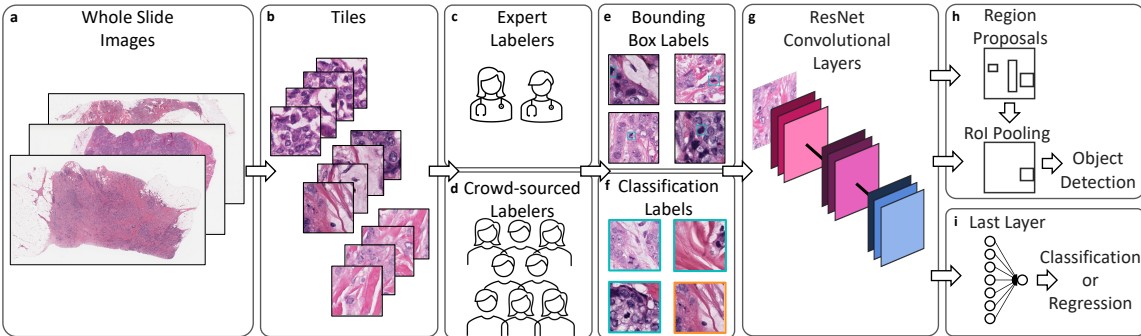

Figure 1: Overview of the analysis pipeline describing the workflow from whole slide image (WSI, a) to prediction (h, i). Each WSI corresponds to a single patient and is divided into tiles (b) for analysis. Two types of labelers assess the tiles, expert (c) and crowd-sourced (d), and provide one of two types of labels: bounding box (e) or binary (f). These labels are then used to train the model, which uses a ResNet backbone (g) which is the input to an object detection model (Faster-RCNN, h) or classification/regression model (i). See Section 2 for more detail.

gram (The Cancer Genome Network, 2012). Each WSI represents a patient biopsy scanned by a digital microscope, typically at multiple magnifications (Fig. 1a). WSIs are then pre-processed to generate tiles, while discarding any tiles which are primarily background (1b). Tiles are passed on to two sets of labelers for two sets of tasks (Fig. 1c-f). The resulting labels are then used as training and testing data for several model classes (Fig. 1g-i). We detail this pipeline and provide necessary background below.

## 2.1. Dataset & Labels

TCGA is a cancer genomics program containing over 20,000 primary cancer and matched normal samples corresponding to 33 cancer types. The data is publicly available. Breast cancer is one of the most common types of cancer and correspondingly is one of the largest sample sizes on TCGA (TCGA BRCA, 1098 cases; The Cancer Genome Network, 2012). In breast cancer, mitotic evaluation is one of the factors used to determine tumor grade. It also has been the cancer used in the aforementioned challenge problems in mitosis detection.

Clinically relevant labels may be associated with the entire WSI, as in diagnosis, or a particular region (tile), as in mitosis detection. Annotations may also be at the pixel level and modeled as a segmentation task. Collecting pixel-level data is expensive, time-consuming, and error prone given the very large image sizes and domain-specific nature of the task. We do not focus on this formulation here but note that it has been addressed in the literature (e.g., Veta et al., 2013; Naylor et al., 2017; Graham et al., 2019), as well as in the context of alternative formulations, e.g. using bounding boxes to learn segmentation maps (Yang et al., 2018).

Unlike in natural image settings, most people are unable to perform annotation tasks, for example, accurately identify tissue and cell structures. This limits but does not prevent efforts to crowd-source annotations (Fig. 1d; Ørting et al., 2020) and increasingly there are companies which facilitate the gathering of crowd-sourced annotations for medical images.

It is much more expensive and time-consuming to acquire expert labels (Fig. 1c) than crowd-sourced labels, however we typically expect expert labels to be more accurate. For the mitotic detection task past work has shown, even amongst experts, there is a considerable amount of label disagreement (Tabata et al., 2019; Bertram et al., 2020). Because of this, labels may be the aggregated result of several labelers, imparting a notion of uncertainty.

## 2.2. Models & Evaluation Metrics

Classification and object detection models are the most commonly used classes of models to perform mitosis detection. We describe each model using two components: feature representation and prediction. Feature representation is largely a function of architecture choice and training data. Some common architectures include ResNet (He et al., 2016), VGG (Simonyan and Zisserman, 2014), and swin-transformer (Liu et al., 2021). Often these representations are pretrained on ImageNet for classification tasks. The last layer of the network can be easily modified adapt to different tasks (predictions), described below.

Classification, regression and object detection models can all ostensibly be used for mitosis detection. Classification follows intuitively from assigning whether a tile has a mitotic event (we refer to a tile with no mitotic events as a "negative sample") and object detection follows intuitively from predicting bounding boxes surrounding mitotic events. The framing as a regression problem comes from considering a continuously valued confidence score, as would result from aggregating the labels of multiple raters. For a more complete description of the types of models, please see the Appendix.

Object detection and classification models are typically evaluated on some set of metrics derived from the confusion matrix, and, in the case of object detection, conditioned on the intersection over union of the predicted and ground truth labels. Because the number of instances is not bounded in the object detection setting, typically the mean (over the population) average (in the image) metric is reported.

## 3. Experimental setup

### 3.1. Dataset & Labels

This work uses labels generated by two groups: crowd-sourced data labelers and expert data labelers. The crowd-sourced data labelers provide annotations via an app-based platform and are described as a mix of medical doctors, professionals, researchers, and students. More detailed demographic information is not provided but labelers are required to "qualify" based on a small set of "gold standard" labeling tasks in mitosis detection. The expert data labelers are a group of three board-certified pathologists (authors DG, HM, MH) and two research scientists (authors CE, SB) familiar with mitosis detection in histopathology data.

In general, we do not expect that it is possible to collect enough expert data to train a successful model because of the time and budget required. Therefore we do not aim to compare the strategies on this axis. Instead, we aim to evaluate the utility of the crowd-sourced data under difference modeling design choices and quantify the differences in time to acquire, concordance, and accuracy as a function of the type of label collected.

Based on an initial user experience experiment, an image magnification of $40\times$ and a tile size of 1,000 pixels square were selected to best view mitotic events and minimize zooming,

Table 1: Summary description of the trained model classes.

| Model Class | Training Label | Negative Tiles | Output |
|---|---|---|---|
| BC | $c \in \{0,1\}$ | ✓ | $\hat{c} \in [0,1]$ |
| Reg | $a \in \mathbb{R}_{>0}$ | | $\hat{a} \in \mathbb{R}$ |
| Reg-NS | $a \in \mathbb{R}_{\geq 0}$ | ✓ | $\hat{a} \in \mathbb{R}$ |
| Det | $\mathbf{b} \in [x_1, y_1, w, h]$ | | $\hat{\mathbf{b}} \in [x_1, y_1, w, h], \hat{p} \in [0,1]$ |
| Det-NS | $\mathbf{b} \in [x_1, y_1, w, h]$ | ✓ | $\hat{\mathbf{b}} \in [x_1, y_1, w, h], \hat{p} \in [0,1]$ |
| Det-WL | $\mathbf{b} \in [x_1, y_1, w, h], a \in \mathbb{R}_{>0}$ | | $\hat{\mathbf{b}} \in [x_1, y_1, w, h], \hat{p} \in [0,1]$ |
| Det-NS-WL | $\mathbf{b} \in [x_1, y_1, w, h], a \in \mathbb{R}_{\geq 0}$ | ✓ | $\hat{\mathbf{b}} \in [x_1, y_1, w, h], \hat{p} \in [0,1]$ |

respectively. Using these specifications, each WSI in this study has approximately 800 to 3,000 1k × 1k tiles. The variance is due to the differing sizes of the biopsy tissues.

The expert group completed two experiments: (1) placing bounding boxes indicating mitotic events on tiles and (2) labeling tiles with binary indicators for the presence or absence of mitosis. We refer to this data as "expert data" throughout. Expert data is aggregated by taking the majority to determine if mitosis present and then averaging the annotations $[x_1, y_1, w, h]$ if the majority agrees mitosis is present. The results of the first experiment are used as the testing data and the results of both experiments are used to compare the time, accuracy, and concordance of the two strategies.

The tiles used in the expert labeler experiment were required to meet two selection criteria chosen to increase the likelihood of discovering mitotic events. First, only high-grade TCGA BRCA WSIs were chosen as high-grade cancer suggests more mitotic events than lower grade (Elston and Ellis, 1991). Second, from those WSIs, tiles were selected by applying an existing mitotic classification model (Dusenberry and Hu, 2018, referred to as IBM model) to all tiles and then picking the top tiles ranked by the confidence score output. The expert labeler dataset consists 4,000 tiles, corresponding to the top 800 tiles from 5 WSIs.

The crowd-sourced group completed one experiment, placing bounding boxes indicating mitotic events on tiles. We refer to this data as "crowd-sourced data" throughout.

For this experiment, bounding boxes are only included in the dataset if at least five labelers have reviewed the tile and at least two labelers have placed overlapping boxes. An agreement score, $a \in [0,1]$, is reported along with each box to capture this information. This data constituted the primary training data. The input to the crowd-sourced data experiment consists of 70 patient BRCA WSIs with various cancer grades and a total of 56,260 1k×1k tiles. For ten of these 70 WSIs, all non-background tiles were chosen. For the remaining WSIs, all tiles with an IBM model confidence score greater than 0.2 and an additional ten percent of tiles with score less than 0.2 were randomly chosen.

### 3.2. Models & Evaluation Metrics

We train seven models using different combinations of input data and prediction tasks (see Table 1). Specifically we select: (1) binary classification (BC), (2) regression with presence-only agreement score data (Reg), (3) regression with agreement score and negative samples (Reg-NS), (4) object detection (Det), (5) object detection with negative samples

(Det-NS), (6) object detection with weighted loss (Det-WL), and (7) object detection with weighted loss and negative samples (Det-NS-WL). Models 1-5 are standard implementations as described in the Appendix. Models 6 and 7 alter the typical cross-entropy loss for the output object detection network by using a weighted cross-entropy where the weight is the corresponding agreement score.

To enable head-to-head comparisons of the different models, we select the true positive rate (TPR) and false positive rate (FPR) as our primary metrics of interest. To do so, we coarsen the predictions from the object detection model by treating each tile with any number predicted boxes as positive and each tile with no predicted boxes as negative.

**Data post-processing**   In all cases, for model training we further tile the 1k × 1k pixel tiles into 250 × 250 pixel tiles. This choice was motivated by the observation that the height and width each mitotic event is typically 30-60 pixels. In choosing a smaller tile size, the object of interest is a larger fraction of the overall image. We use a step size of 125 pixels such that each tile overlaps with its neighbors. Using this procedure, each 1k × 1k pixel tile results in 49 250 × 250 pixel sub-tiles. Training, validation, and testing data splits are done respecting the source WSI such that one patient's data does not span the split.

**Label post-processing**   Although our goal is to explore the interactions between data collection and model through classification, regression, and object detection in the context of mitotic detection, collecting each of these datasets is cost prohibitive and therefore we gathered the most detailed data, bounding boxes, and applied coarsening techniques to use the data in the classification and regression settings. For the regression task we use the agreement score ($a$) of the tile, weighted by the fraction ($r$) of the bounding box contained in the tile, combined using a weighted sum in the case of multiple events ($N$) per tile ($t$), $s_t = \sum_{i=1}^{N} r_i \cdot a_i$, where $r, a \in [0, 1]$. These scores are used directly for the regression task. For the classification task, we assign tiles with a non-zero agreement score to be in the mitotic class and those with a zero agreement score to be classed as not mitotic.

**Training Setup**   We describe our training data on the basis of unique mitotic events and tiles without any mitotic events (negative samples). Because of the nature of the problem, many tiles do not have any mitotic events and due to our sub-tiling strategy, most mitotic events appear on multiple tiles. Therefore, we assign a unique identifier to each mitotic event. To obtain a data point (tile) from the dataset, we first sample a mitotic event ID. Then we sample a tile from the pool of tiles that contain the sampled mitotic event. Using this two-step process (referred to as "sampling strategy"), we clearly define the number of mitotic events used during training and introduce an augmentation as the mitotic event is effectively moved around the tile as a result of the different crops. Binary classification models are trained with negative samples (1:1 ratio) and regression and object detection models are trained with various ratios (0:1 to 10:1 ratios).

All models are implemented in PyTorch (Paszke et al., 2019) using torchvision. We leverage built-in functions and pipelines which can easily be reproduced (specific code for this study will not be released). Please see the Appendix for additional training details.

Table 2: Labeled data summary

| Partition | Type | # Source WSI | # Mitotic Events |
|---|---|---|---|
| Train | Crowd | 61 | 42,203 |
| Validation | Crowd | 9 | 10,484 |
| Test | Expert | 5 | 612 |

The number of tiles per WSI varies, see Section 3.1

Table 3: Label acquisition comparison

| | Fleiss Kappa | Time (sec) | Mean Accuracy |
|---|---|---|---|
| Binary Classification | 0.709 | 2.7 | 89% |
| Bounding Box Placement | 0.618 | 39 | 88% |

Table 4: Summary of model performance on the test dataset. The model threshold is reported where relevant and NS refers to the ratio of negative to positive samples in the training data.

| Model | BC | Reg | Reg | Reg | Reg | Reg | Det | Det | Det-WL | Det-WL |
|---|---|---|---|---|---|---|---|---|---|---|
| Threshold | 0.5 | 0.6 | 0.4 | 0.15 | 0.2 | 0.25 | 0.95 | 0.95 | 0.95 | 0.95 |
| NS | 0:1 | 0:1 | 1:1 | 2:1 | 4:1 | 10:1 | 0:1 | 1:1 | 0:1 | 1:1 |
| TPR % ↑ | 91.4 | 75.8 | 86.7 | **93.8** | **89.1** | 79.8 | 90.6 | 67.5 | 93.2 | 80.5 |
| FPR % ↓ | 40.8 | 53.3 | 26.0 | 42.3 | **24.5** | **12.4** | 30.6 | 13.3 | 43.5 | 25.6 |

## 4. Results

### 4.1. Labeling Results

The labeled data are summarized in Table 2. In the expert data, only 506 out of 4,000 tiles are mitotic for a total of 612 mitotic events. In the crowd-sourced data, 34,987 out of 56,260 tiles are labeled mitotic with a total of 52,687 mitotic events.

Table 3 compares the two labeling tasks amongst the expert labelers. In both settings, moderate concordance is achieved, although concordance is higher in the binary task, as measured by Fleiss kappa. Unsurprisingly, tile-level annotations are faster to acquire. Once tile size is accounted for, the time to acquire the same number of annotated pixels is approximately the same: there are 16 non-overlapping $250 \times 250$ pixel tiles per each $1k \times 1k$ tile and $2.7 \text{ sec} \times 16 \approx 40 \text{ sec}$. However, the diversity of tile-level annotations collected can be greater given the same time budget by selecting a more spatially diverse set. We also report accuracy, where ground truth is based on the majority expert vote, and find no appreciable difference in the two tasks.

As the crowd-sourced labelers have less experience, we generally expect their labels to have more errors. We observe that the rate of mitotic events per WSI is much higher among crowd-sourced data. To assess the quality of the crowd-source label results, we asked the experts to assess the predictions. We randomly selected 500 tiles from the crowd-sourced data and asked the experts to confirm/reject the box as well as add any missing boxes. The experts placed no new boxes and confirmed $\approx 85\%$ of the bounding boxes.

### 4.2. Modeling Results

The test results for the seven model classes are summarized in Table 4. Regression models with negative samples achieved the highest true positive rate and lowest false positive rate,

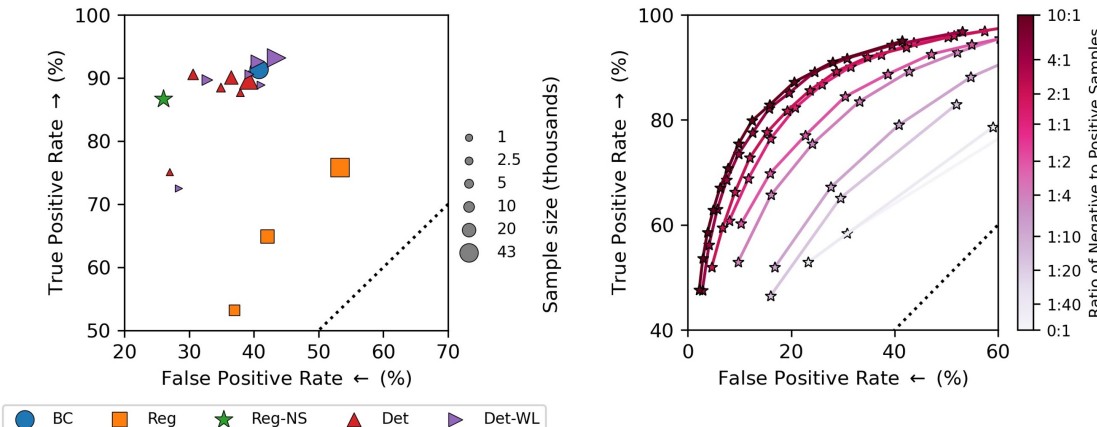

Figure 2: **Left**: TPR and FPR as a function of training samples. **Right**: Performance improves with more negative samples in almost all instances. TPR and FPR for various Reg-NS models using all 43K mitotic events. Each color represents a ratio of negative to positives samples, from 0:1 to 1:1. Each star represents a threshold from 0.15 to 0.60 with a step size of 0.05.

although not in the same model. Taking both metrics into account, regression with negative samples (Reg-NS) using a 4:1 negative to positive sample ratio had the best performance. Overall, we observe benefit from including negative samples and the agreement score. When looking at how the performance changes as a function of dataset size, we generally observe both TPR and FPR increasing together as dataset size increases (see Fig. 2, App. Tab. 5).

Depending on the particular setting, the relative importance of the TPR and FPR performance will vary. This trade-off can be explicitly addressed by setting prediction thresholds. We investigated the impact of this threshold, along with different levels of negatives samples, on performance (see Fig. 2 and Appendix Fig. 3, Tables 6-9). It is important to acknowledge that the expert data testing set has a large class imbalance, as is expected. The dataset consists of 196,000 250 × 250 tiles: 1,835 tiles have at least one mitotic event and 194,165 tiles have no mitotic events. Rather small changes in the FPR greatly change the number of tiles that are predicted positive.

For Det and Det-WL models, we also explored loading the ResNet50 backbone obtained from the regression task to see whether a pathology-specific pre-training task could improve the performance of the detection model. However, the performance was worse than our normal setting starting from PyTorch pre-trained weights using ImageNet (see Appendix Table 8). We hypothesize that this is due to the small number of samples used to train the regression model as compared to the size of ImageNet.

We also performed comparisons of aggregating all tiles with mitotic events into the training dataset, in contrast to the sampling strategy applied, and found similar to worse performance depending on the exact setting. However in all settings, the naive aggregation greatly increased the computational cost, thereby motivating our sampling approach.

## 5. Discussion

When given the opportunity to consider the full ML pipeline from label collection to model training, there will be many factors to consider such as labeler availability, cost, and quality, and desired model output. Because of the diversity of settings, hard-and-fast rules are not possible, however we present recommendations based on our findings.

Our analysis shows the utility of crowd-sourced data along with the importance of expert data and therefore, we recommend gathering both. The crowd-sourced data was relatively inexpensive, though noisy, and the expert data was critical for performance evaluation. In practice, we recommend using some expert data to fine-tune models after training with crowd-sourced data. We also stress the importance of multiple reviews per instance. Our concordance analysis highlights the challenging nature of the task. Leveraging metrics such as the agreement score largely improved performance.

In the context of framing the problem as a regression vs object detection problem, we admit initial surprise by the success of the regression models. *A priori* we assumed that the signal-to-noise ratio would be much lower for tile-level annotations and that this would negatively impact model performance. However, particularly when comparing the Reg and Reg-NS results, it becomes clear that negative samples are valuable for improving model performance. Most tiles do not have mitotic events and explicitly using the negative samples when calculating the loss function is important. This is a notable difference with object detection in natural images, where negative samples are typically rare.

Promising directions for future work are more sophisticated incorporation of negative instances and uncertainty during training. And although the regression models had strong performance, the object detection model provides the additional information of the box, which inherently renders the model more interpretable.

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

## Appendix A. Further description of training tasks

**Classification** A mitosis detection classification model takes a tile as input and outputs a continuous value $\hat{c} \in [0,1]$ which can then be thresholded to generate a binary indicator for the presence of mitosis (Fig. 1i). In a well calibrated model, it is expected that tiles resulting in larger predicted values of $\hat{c}$ are more likely to contain mitotic events. If the total count of mitotic events is the desired output, multi-class classification can also be considered. Typically, classification models are trained using cross-entropy loss. The work of Dusenberry and Hu (2018) is an example of binary classification for mitotic detection. Albarqouni et al. (2016) also uses classification, and notably considers different sources of labeling.

**Regression** Although mitosis is most typically cast as a binary indicator, the presence of mitosis can be recast as a continuous representation, e.g. by aggregating the labeling of multiple raters by average the binary labels. The resulting regression model takes a tile as input and outputs a confidence score, $\hat{a} \in \mathbb{R}$, by replacing the last layer of the binary classification architecture with a layer which generates a scalar output (Fig. 1i). The model is trained using the mean-squared error loss.

**Object detection** Object detection models are trained using a vector which describes the bounding box, i.e., coordinates corresponding to specific regions on the tile and the corresponding box size, e.g., $[x_1, y_1, w, h]$. The detection model takes a tile as input and outputs (possibly zero) bounding boxes and an associated score (Fig. 1h). The Faster-RCNN ((Ren et al., 2015)) is one of the most widely used object detection models for natural images. It utilizes the representation from CNN (ResNet50) backbone to predict the location and class of bounding boxes. The model is trained with smoothed L1 loss for positions of the box and cross-entropy loss for the objectiveness and the class of bounding box. Compared to regression models, object detection models provide the extra information of the exact locations of mitotic events in the tile. This inherently enables a more interpretable output as compared to classification and regression, as labelers typically interact with data of this form. Several studies have considered Faster-RCNNs and related RCNN methods for mitosis detection (e.g., Li et al., 2018; Lei et al., 2019; Sebai et al., 2020b; Sohail et al., 2021).

## Appendix B. Additional training details

Based on initial tuning experiments, for the BC, Reg, and Reg-NS models, we choose a batch size of 32, learning rate of 0.005, and used the Adam optimizer ((Kingma and Ba, 2014)) with a step learning rate scheduler ($\gamma = 0.1$ and step size of 15). Models are trained for 60 epochs. For all Det(-) models, we choose a batch size of 16, learning rate of 0.005, and used the SGD optimizer with a step learning rate scheduler ($\gamma = 0.1$ and step size of 20). Models are trained for 30 epochs. For all models, during training, we applied random horizontal flip and random photo-metric distortion from torchvision helper function as image augmentation. For BC, Reg, and Reg-NS, we chose to start with torchvision's pre-trained ResNet50 and modified the last layer. For BC, Reg, and Reg-NS, the best model is saved when the loss is lowest on the validation set. For all Det(-), we used the pre-trained fastrcnn-resnet50-fpn model with built-in object detection helper functions to evaluate trained model on validation set and saved the model with the best performance. We also investigated the use of validation data containing negative samples for the selection of detection models (Det and Det-WL) and found that this decreased performance.

## Appendix C. Additional model performance data

Please see Tables 5, 6, 7, and 8 for detailed results on the effect of training data size, negative sample size and threshold and ResNet backbone, respectively.

Table 5: Detailed data of Fig 2 of model performance with different training samples.

| #Samples
Model | 1K | 2.5K | 5K | 10K | 20K | 42K (All) | All NS |
|---|---|---|---|---|---|---|---|
| BC (0.5) | | | | | | | 91.4/40.8 |
| Reg (0.6) | 100/100 | 89.9/83.2 | 99.6/98 | 53.2/37 | 64.9/42.1 | 75.8/53.3 | |
| Reg-NS (0.35) | | | | | | | 90.1/31.5 |
| Det | 75.1/27 | 87.7/37.9 | 88.5/34.9 | 90.6/30.6 | 90.1/36.5 | 89.7/39.2 | |
| Det-WL | 72.5/28.4 | 88.9/41.1 | 90.6/39.3 | 89.7/32.8 | 92.6/40.6 | 93.2/43.5 | |

Table 6: Detailed data of Fig 2 of model performance with different negative samples and different score threshold.

| #NS
Model | 0 | 1K | 2.5K | 5K | 10K | 20K | 42K (All) |
|---|---|---|---|---|---|---|---|
| Reg (0.6) | 75.8/53.3 | | | | | | |
| Reg (0.7) | 6.9/3.6 | | | | | | |
| Reg-NS (0.15) | | | | | | 97.7/72.5 | 96.9/57.4 |
| Reg-NS (0.2) | | | | | | 96.7/67.4 | 95.7/50.3 |
| Reg-NS (0.25) | | | | | 96.8/67.5 | 95.7/61.6 | 94.7/43.7 |
| Reg-NS (0.3) | | | | | 95.5/60.4 | 94.3/54.9 | 92.4/37.4 |
| Reg-NS (0.35) | | | | | 92.8/52.1 | 92.5/47.1 | 90.1/31.5 |
| Reg-NS (0.4) | | | | 93.8/67.8 | 89.2/42.8 | 88.7/38.7 | 86.7/26 |
| Reg-NS (0.45) | | | | 88.1/54.7 | 83.5/33.2 | 84.4/30.4 | 82.3/20.8 |
| Reg-NS (0.5) | | | 82.9/51.9 | 79/40.8 | 75.4/24.1 | 77/22.7 | 76.4/16 |
| Reg-NS (0.55) | | 78.6/59 | 65.1/29.5 | 67.2/27.7 | 65.7/16.1 | 69.8/15.9 | 68.8/11.7 |
| Reg-NS (0.6) | | 52.9/23.3 | 46.5/16 | 51.9/16.8 | 52.9/9.7 | 60.2/10.3 | 60.8/8 |
| BC | | | | | | | 91.4/40.8 |
| Det | 89.7/39.2 | | | | | | |
| Det-WL | 93.2/43.5 | | | | | | |

Table 7: Detailed data of Fig 2 of model performance with different negative samples and different score threshold.

| Model \ #NS | 1:1 (All) | 2:1 | 4:1 | 10:1 |
|---|---|---|---|---|
| Reg (0.6) | 75.8/53.3 | | | |
| Reg (0.7) | 6.9/3.6 | | | |
| Reg-NS (0.15) | 96.9/57.4 | 93.8/42.3 | 91.7/30.8 | 87.2/20.6 |
| Reg-NS (0.2) | 95.7/50.3 | 91.9/34.7 | 89.1/24.5 | 82.8/15.8 |
| Reg-NS (0.25) | 94.7/43.7 | 89.2/28.7 | 85.1/19.6 | 79.8/12.4 |
| Reg-NS (0.3) | 92.4/37.4 | 85.6/23.6 | 82.1/15.7 | 75.4/9.8 |
| Reg-NS (0.35) | 90.1/31.5 | 81.7/19.2 | 77.5/12.5 | 70.7/7.8 |
| Reg-NS (0.4) | 86.7/26 | 77.6/15.3 | 73.5/9.8 | 67.0/6.2 |
| Reg-NS (0.45) | 82.3/20.8 | 72.8/12.0 | 68.5/7.5 | 62.8/4.9 |
| Reg-NS (0.5) | 76.4/16 | 66.2/9.1 | 63.0/5.7 | 58.6/3.8 |
| Reg-NS (0.55) | 68.8/11.7 | 59.4/6.7 | 56.2/4.1 | 53.6/2.9 |
| Reg-NS (0.6) | 60.8/8 | 51.9/4.6 | 47.5/2.8 | 47.6/2.2 |
| BC | 91.4/40.8 | | | |
| Det-NS | 67.4/13.3 | 63.5/9.4 | 36.9/2.7 | 16.7/2.2 |
| Det-WL-NS | 80.5/25.6 | 76.0/24.4 | 76.0/26.4 | 66.5/24.9 |

Table 8: Performance of Detection model by loading ResNet50 backbone weights pretrained on Regression task

| Det Model \ Backbone | Reg1K | Reg2.5K | Reg5K | Reg10K | Reg20K | Reg42K(All) | ImageNet |
|---|---|---|---|---|---|---|---|
| Det 1K | 3.1/2.4 | 3.6/2.5 | 1.9/1.5 | 4.4/4.2 | 6.9/3.3 | 2/0.7 | 75.1/27 |
| Det 2.5K | 11/8.5 | 20.2/13.5 | 11.6/7.3 | 15.2/10.8 | 24.8/13.5 | 15/7.8 | 87.7/37.9 |
| Det 5K | 17.9/12.1 | 28.7/18.7 | 19.7/9.9 | 20.8/12.5 | 30.1/15.1 | 19.9/12.3 | 88.5/34.9 |
| Det 10K | 24.4/17.9 | 32.9/18.6 | 21.5/11.5 | 27.2/15.7 | 29.6/15.3 | 22.7/14.3 | 90.6/36.6 |
| Det 20K | 26/17.8 | 47.5/23.1 | 31.9/20 | 43.6/21.4 | 35/18.7 | 13.7/7.1 | 90.1/36.5 |
| Det 42K(All) | 41.9/18.1 | 58.7/22.1 | 28.6/13.8 | 53.4/22.9 | 34.4/17.2 | 26.6/17.3 | 89.7/39.2 |
| Det-WL 1K | 5.6/4.1 | 2.7/4.8 | 3.1/1.2 | 7/6.3 | 10.9/5.6 | 3.1/1.9 | 72.5/28.4 |
| Det-WL 10K | 20.2/15.6 | 27.5/18.2 | 14.3/7.4 | 28.2/17.4 | 35.1/18.9 | 18.4/12.1 | 89.7/32.8 |
| Det-WL 20K | 34.4/23.5 | 43.3/26.8 | 24/13.5 | 31.7/17.5 | 39.7/21.6 | 32.1/22.2 | 92.6/40.6 |
| Det-WL 42K(All) | 39.5/19.1 | 57.4/24.9 | 30/16.80 | 37.5/14.7 | 34.4/16.7 | 26.5/14.1 | 93.2/43.5 |

Table 9: Detailed data of Fig. 3 of model performance with different prediction thresholds for the detection models.

| Model
Threshold | Det | Det-NS | Det-WL | Det-WL-NS |
|---|---|---|---|---|
| 0.925 | 93.5/47.1 | 75.2/19.5 | 95.9/51.2 | 86.0/33.1 |
| 0.95 | 90.6/30.6 | 67.6/30.6 | 93.2/43.5 | 80.5/25.6 |
| 0.975 | 82.2/26.4 | 49.3/5.3 | 86.7/30.2 | 63.4/13.2 |

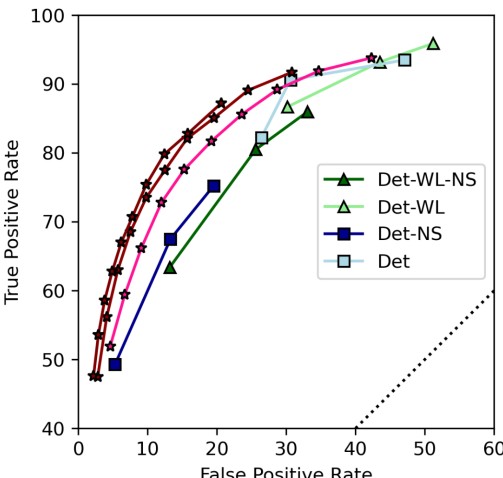

Figure 3: The true positive and false positive rates for various models using different prediction thresholds. The regression results (stars) are reproduced from Fig. 2 in the text and include only the top performing models. The detection models have greater sensitivity to threshold (step size of 0.025 as compared to 0.05 in the regression case) and consistently underperform or perform no better than the regression models.

