# OpenReview forum: "Considerations for data acquisition and modeling strategies: Mitosis detection in computational pathology"
_MIDL.io/2023/Conference — MIDL 2023 Poster_

### Official Review · Reviewer_sPoH · 2023-02-01

**Confidence:** 3
**Preliminary Rating:** 4
**Recommendation:** Oral, Poster

**Summary:**

The paper compares classification and object detection method with respect to labeling and algorithm performance for the task of mitosis detection on pathological images. While the object detection setup is quite straight forward, the classification is broken down to tiled classification of small patches. Overall, the results show that classification can be competitive with object detection for this task.

**Strengths:**

+ The paper is nicely written
+ The paper considers both aspects, labeling and automated classification
+ The experiments are nicely performed
+ The results are nice and interesting and a little surprising (not sure whether that actually counts as a strength)
+ Not many papers consider such aspects

**Weaknesses:**

- Methodological there is not much actual development
- I feel like detection methods do not come across as good in the text as they seem in Fig 2. and the time benefits are a bit misleading, since as far as I understood, due to the tiling, there are none (if tiling is used and not a more sparse sampling...)
- No code will be released

**Deanonymize Review:**

no

**Paper Type:**

both

**Questions To Address In The Rebuttal:**

I feel like the time savings (if there are any) should be discussed in more detail and what benefits the actual approach has (since it becomes a little more ambiguous due to the tiling).
Further I think it could be good to discuss the added choice of threshold the regression/classification methods have, and how much that can actually influence the results (and if/why that can be feature), and if sth similar can be done for detection (where boxes usually also get confidence scores...).

---

### Official Review · Reviewer_YPn7 · 2023-02-01

**Confidence:** 5
**Preliminary Rating:** 2

**Summary:**

This paper aims to investigate into the implication of data collection strategies for the task of mitosis detection. To be specific, the author investigates the use of 1) expert and crowd-sourced labelers, 2) the impact of aggregated vs single labels, and 3) the framing of the problem: classification or regression.

**Strengths:**

This paper provides a detailed description of the label collection process. The efforts to compare the cost and quality of different label sources could be valuable for the practitioner in this field.

This paper further gives some take-away information: 1) both types of labels are important. 2) negative samples are important. They further suggest future directions.

**Weaknesses:**

it is a little too ambitious for the 8-page paper to answer the question from so many aspects 1) different types of label: fine-grained or high-level labels 2) different sources of labels: expert, crowd-sourcing 3) different label processing strategies: aggregation, sampling etc; 4) different task: regression or classification. To this end, insufficient experiments and analyses are provided, making it pretty hard to draw any strong conclusion.

1) the parts labeling results and modeling results in the experiments section do not seem to be bridged well. Those two look like two separate parts. How do the labeling results influence the modeling results?

2) all the factors are compared in the same table or figure, not in a controlled setting. The main messages in the conclusion part could not be bridged with the analysis or evidence in the experiment clearly. For example, the author mentioned the crowd-sourced data was noisy and the expert data are of higher quality, this is intuitive, but the conclusion that using both is important does not seem to be well supported by the experiment result.

3) the author mentioned the "sampling" strategy instead of aggregating, however, when I searched for the "sampling" in the paper, I could not find any definition for that





**Deanonymize Review:**

no

**Detailed Comments:**

See above

**Paper Type:**

validation/application paper

**Questions To Address In The Rebuttal:**

1. clearly summarize the main conclusion of this paper, with clear evidence from either the existing results in the paper or the new results provided during the rebuttal.

2. organize the experiments in a controlled setting so that it is clear what factor the author mainly wants to compare.

3. provide a clear analysis of the experiments rather than simply describing them.

---

### Official Review · Reviewer_ff56 · 2023-02-03

**Confidence:** 5
**Preliminary Rating:** 4
**Recommendation:** Poster

**Summary:**

The paper studies the implication of label collection strategies and their relationship with ML models developed using them, using mitosis detection in pathology as a case study. The paper creates an experimental setup involving different labelers (expert vs crowd sourced), labeling strategies (bounding box vs binary labels) which is used for training different model formulations (regression vs classification). An analysis of the labeling and modeling are presented and learnings are summarized. The experimental study is well designed, the analysis is well done, and findings presented are useful for both research and applications in computational pathology.

**Strengths:**

1. The study is interesting, well motivated and relevant. Label noise, its effect on model formulation and performance, as well as other tradeoffs associated with label collection (time, cost, quality) are important areas of empirical research in the community.
2. The paper is clearly written and the experimental details are very well described.
3. Labeling results were expected, but insightful to see quantitatively evaluated. The nuance around type of labels, speed of acquisition, diversity of samples and effect on model performance is valuable for practitioners.
4. In modeling, the significance of negatives, the role of sampling tiles during training, and the comparison across models, are all useful baselines for future work in mitosis detection.

**Weaknesses:**

1. The description for crowd-sourced data and expert data is missing information about the labeler's background. It is important contextual information for the study to define what kind of labelers are included in each group.
2. I'm not sure if the recommendation to fine-tune with expert labels after training with crowd-sourced labels follows from the experiment. The experiments do not study the effect of label noise on the final test performance. I agree that the 20% false positives do indicate that clean labels from experts are needed for the final evaluation.
3. The paper is also missing the ablating the effect of using lesser number of labels per instance even though it's mentioned in the final discussion. It would have been interesting to see if using fewer labelers leads to poorer models.

**Deanonymize Review:**

no

**Paper Type:**

validation/application paper

**Questions To Address In The Rebuttal:**

1. Would like to see the details around labeler group characterization.
2. It would be helpful if the authors explain why the recommendation of fine-tuning with expert labels and using multiple labelers per instance is justified based on the experimental findings.
3. Suggestion to add some missing references on previous work which discusses the labeling aspects for pathology ML models [1, 2, 3].


[1] Albarqouni, S., Baur, C., Achilles, F., Belagiannis, V., Demirci, S., & Navab, N. (2016). Aggnet: deep learning from crowds for mitosis detection in breast cancer histology images. IEEE transactions on medical imaging, 35(5), 1313-1321.

[2] Bertram, C. A., Veta, M., Marzahl, C., Stathonikos, N., Maier, A., Klopfleisch, R., & Aubreville, M. (2020). Are pathologist-defined labels reproducible? Comparison of the TUPAC16 mitotic figure dataset with an alternative set of labels. In Interpretable and Annotation-Efficient Learning for Medical Image Computing: Third International Workshop, iMIMIC 2020, Second International Workshop, MIL3ID 2020, and 5th International Workshop, LABELS 2020, Held in Conjunction with MICCAI 2020, Lima, Peru, October 4–8, 2020, Proceedings 3 (pp. 204-213). Springer International Publishing.

[3] Javed, S. A., Juyal, D., Shanis, Z., Chakraborty, S., Pokkalla, H., & Prakash, A. (2022). Rethinking machine learning model evaluation in pathology. arXiv preprint arXiv:2204.05205.

---

### Meta-Review · Area_Chair_XHmT · 2023-02-26

**Recommendation:** Accept (Poster)
**Confidence:** 4

**Metareview:**

This is a validation/application paper, which considers a practically interesting setting. It investigates the use of both expert (clean) and crowd-sourced (potentially noisy) labels and the impact of aggregated vs single labels, in the context of mitosis detection (both classification and object detection). Overall, the reviewers agree that this experimental setting is interesting, well motivated and of value to practitioners in MIDL. The  results are also interesting and demonstrate the value of crowd-sourced labels, the importance of uncertainty quantification, and the usefulness  of negative samples. The paper is well-written, the experiments are well conducted and the authors provided convincing answers during the discussion. I also like the recommendation to fine-tune with expert labels after training with crowd-sourced labels (although not supported with experiments). This is, in a sense, in line with the paradigm of foundation models in learning (learn from large-scale unlabeled data and fintune one a few labeled samples). Overall, I feel this type of experimental studies, quite uncommon in the literature, could trigger nice discussions at MIDL. Hence, I recommend acceptance.